# Rapid sex-specific adaptation to high temperature in *Drosophila*

**Sheng-Kai Hsu[1,2], Ana Marija Jakšić[1,2†], Viola Nolte[1], Manolis Lirakis[1,2], Robert Kofler[1], Neda Barghi[1], Elisabetta Versace[3], Christian Schlötterer[1]\***

[1]Institut für Populationsgenetik, Vetmeduni Vienna, Vienna, Austria; [2]Vienna Graduate School of Population Genetics, Vetmeduni Vienna, Vienna, Austria; [3]Department of Biological and Experimental Psychology, Queen Mary University of London, London, United Kingdom

**Abstract** The pervasive occurrence of sexual dimorphism demonstrates different adaptive strategies of males and females. While different reproductive strategies of the two sexes are well-characterized, very little is known about differential functional requirements of males and females in their natural habitats. Here, we study the impact environmental change on the selection response in both sexes. Exposing replicated *Drosophila* populations to a novel temperature regime, we demonstrate sex-specific changes in gene expression, metabolic and behavioral phenotypes in less than 100 generations. This indicates not only different functional requirements of both sexes in the new environment but also rapid sex-specific adaptation. Supported by computer simulations we propose that altered sex-biased gene regulation from standing genetic variation, rather than new mutations, is the driver of rapid sex-specific adaptation. Our discovery of environmentally driven divergent functional requirements of males and females has important implications-possibly even for gender aware medical treatments.

**\*For correspondence:**
Christian.Schloetterer@
vetmeduni.ac.at

**Present address:** †Department for Molecular Biology and Genetics, Cornell University, New York, United States

**Competing interests:** The authors declare that no competing interests exist.

## Introduction

The ubiquity of sexual dimorphism in dioecious organisms reflects the discordant selection pressure driven by divergent reproductive roles of males and females (*Chapman, 2006*). For instance, males typically evolve to increase their mating frequency and success of fertilization, while females benefit from better resource allocation to their offspring (*Brengdahl et al., 2018*; *Civetta and Clark, 2000*; *Friberg and Arnqvist, 2003*). Often, such differential requirements of males and females results in sexual conflict, preventing males and females to reach sex-specific trait optima (*Bonduriansky and Chenoweth, 2009*; *Lande, 1980*; *Mank, 2017a*; *Rice, 1992*). Based on the widespread sexual dimorphism, several models for the evolution of sexual dimorphism from a largely shared genome have been proposed (*Barson et al., 2015*; *Day and Bonduriansky, 2004*; *Mank, 2017b*; *Parsch and Ellegren, 2013*; *Pennell and Morrow, 2013*; *Rice, 1984*; *Telonis-Scott et al., 2009*). One implicit assumption of these studies is that stable sex-specific fitness landscapes are persisting over long evolutionary time scales. However, ecological changes, such as environmental fluctuations, occur at high rates (*Reznick and Ghalambor, 2001*). If such environmental factors affect the sex-specific fitness landscapes, sudden ecological changes may impose selection for novel/altered sexual dimorphism in a population (*Camus et al., 2019*).

To date, limited attention has been given to the evolutionary dynamics of sex differences in response to changing environments. The clinal variation of sexual dimorphism for a small number of phenotypes (*Blanckenhorn et al., 2006*; *Chenoweth et al., 2008*) and gene expression (*Allen et al., 2017*; *Hutter et al., 2008*) in *Drosophila* suggests that sex-specific adaptation in response to environmental heterogeneity is not uncommon. When the requirements of males and females differ in an environment-specific manner, the adaptive response is contingent on the availability of

**eLife digest** Male and female animals of the same species sometimes differ in appearance and sexual behavior, a phenomenon known as sexual dimorphism. Both sexes share most of the same genes, but differences can emerge because of the way these are read by cells to create proteins – a process called gene expression. For instance, certain genes can be more expressed in males than in females, and vice-versa.

Most studies into the emergence of sexual dimorphism have taken place in stable environments with few changes in climate or other factors. Therefore, the potential impact of environmental changes on sexual dimorphism has been largely overlooked.

Here, Hsu et al. used genetic and computational approaches to investigate whether male and female fruit flies adapt differently to a new, hotter environment over several generations. The experiment showed that, after only 100 generations, the way that 60% of all genes were expressed evolved in a different direction in the two sexes. This led to differences in how the males and females made and broke down fat molecules, and in how their neurons operated. These expression changes also translated in differences for high-level biological processes. For instance, animals in the new settings ended up behaving differently, with the males at the end of the experiment spending more time chasing females than the ancestral flies.

These findings demonstrate that male and female fruit flies adapt many biological processes (including metabolism and behaviors) differently to cope with changes in their environment, and that many different genes support these sex-specific adaptations. Ultimately, the work by Hsu et al. may inform medical strategies that take into account interactions between the patient's sex and their environment.

segregating variants with sex-specific or sex-biased effects. Without the corresponding variants, sex-specific adaptation requires new mutations, resulting in slow evolutionary responses. Here, we use experimental evolution for direct experimental evidence that sex-specific adaptation can be triggered by a rapid environmental shift within a few generations.

## Results and discussion

### Distinct phenotypic changes of females and males in a novel environment

We explored the phenotypic evolution of males and females by studying gene expression because many of these molecular phenotypes can be scored with a high precision at moderate costs. Furthermore, in contrast to high-level phenotypes, which are typically selected on a priori criteria, the analysis of gene expression is unbiased. We measured gene expression of 10 replicate populations which evolved independently for more than 100 generations in a simple and well-controlled high-temperature selection regime (*Barghi et al., 2019*). In each sex, we screened for genes with parallel changes in expression across the replicated evolved populations compared to their same-sex ancestors. After accounting for allometric changes during evolution (see Materials and methods), we identified 2366 and 4151 genes (25% and 44% of all expressed genes, N = 9,457) showing evolutionary responses in males and females respectively (FDR < 0.05; *Supplementary file 1* and *Figure 1—figure supplement 1*). The evolution in gene expression was vastly different between the sexes, resulting in almost uncorrelated gene expression changes (*Figure 1a*). Only 760 genes (14%; 469 up-regulated and 291 down-regulated) evolved concordantly in both sexes. 1295 genes (24%) changed expression specifically in males (657 up-regulated and 638 down-regulated) and 3080 genes (57%) evolved in females only (1877 up-regulated and 1203 down-regulated). Interestingly, 311 genes (6%) displayed divergent responses to selection in the two sexes (*Figure 1b*). The discordant gene expression evolution of males and females indicates different functional requirements in the novel environment.

To determine the diverged functional requirements of males and females in the new environment, we tested for enrichment of gene ontology (GO) terms and tissue-specific expression (*Figure 1c and d*, *Supplementary files 2* and *3*). We found a striking pattern of enrichment that suggested sex-specific evolution of fatty acid metabolism in both the GO term and tissue-specific enrichment analyses.

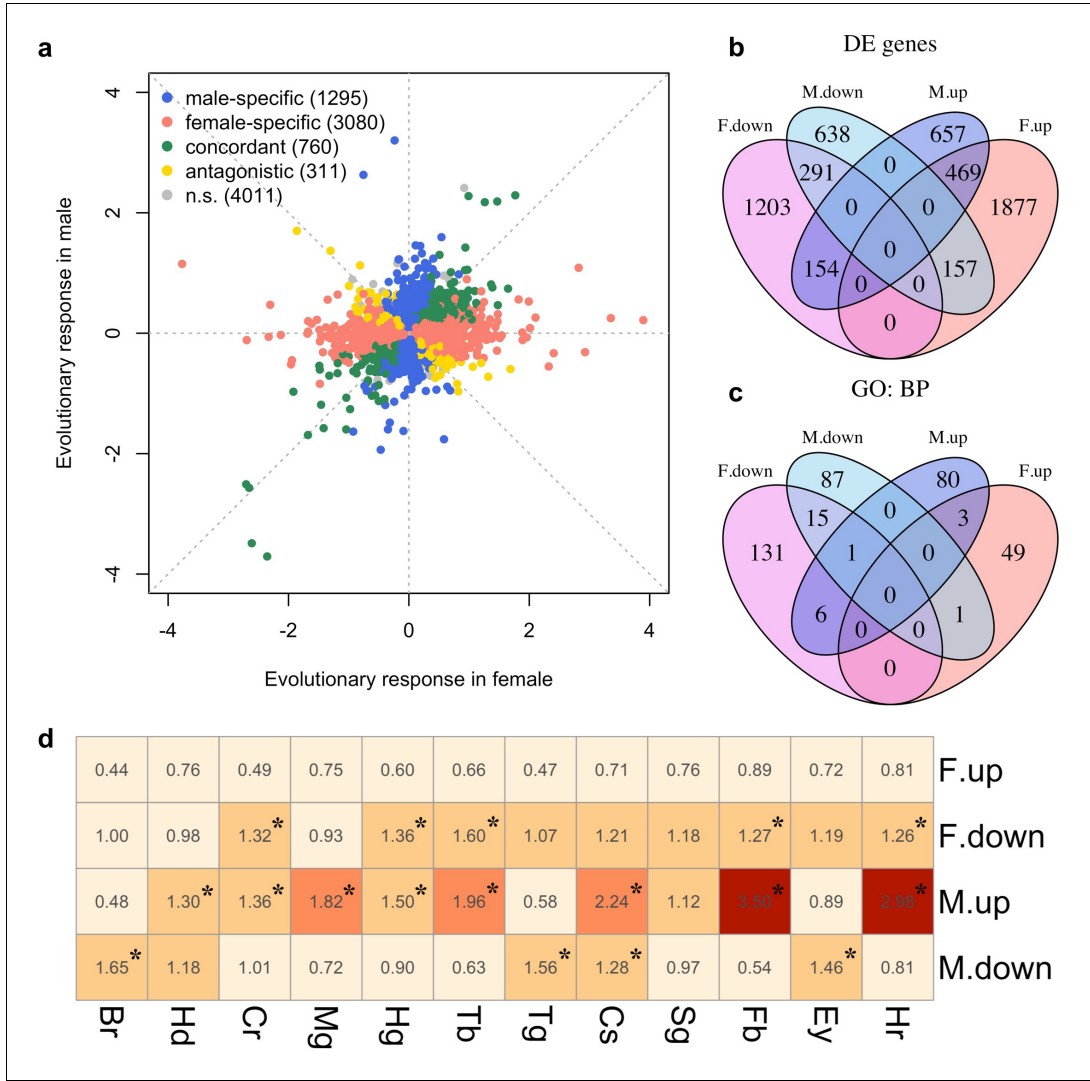

**Figure 1.** Sex-specific gene expression evolution adapting to a high temperature. (a) Evolution of gene expression in females (x axis) and males (y axis). The evolutionary changes of all expressed genes are shown on $\log_2$ scale. Genes showing different patterns of evolution are highlighted in different colors. (b) The majority of the genes with significant expression changes is sex-specific. Venn diagram showing the number of genes with significantly different gene expression patterns (DE: Differential Expression; M.up/F.up: males/females evolved higher gene expression, M.down/F.down: males/females evolved lower gene expression). (c) Genes with evolved expression changes in males and females are involved in nearly mutually exclusive sets of biological processes. Venn diagram of sets of GO (Gene ontology: biological processes) terms enriched by the genes changing their expression for each direction in each sex (i.e. four sets of candidate genes: up/down-regulation in males/females). For instance, there are only three biological processes repeatedly found among the 90 and 53 processes involving up-regulated genes in males and females respectively. (d) The tissue enrichment of genes significantly evolving for either direction in males and females (Br-brain, Hd-head, Cr-crop, Mg-midgut, Hg-hindgut, Tb-malpighian tubule, Tg-thoracoabdominal ganglion, Cs-carcass, Sg-salivary gland, Fb-fat body, Ey-eye and Hr-heart). Each cell represents the result of a Fisher's exact test. The colors and numbers denote the magnitude of odds ratio and statistical significance (FDR < 0.05) is indicated with *. Consistent with GO enrichment results, gene expression evolution in males and females may occur in different tissues.

The online version of this article includes the following figure supplement(s) for figure 1:

**Figure supplement 1.** Parallel responses of adaptive genes across replicates.

**Figure supplement 2.** Evolution of sexual dimorphism.

Genes highly expressed in fat body tissue were over-represented among the 1280 genes with upregulation in males, but over-represented among the 1648 genes with downregulation in females (FET, FDR < 0.01 in both tests, *Figure 1d* and *Supplementary file 3*). GO enrichment analysis of genes with male-specific upregulation further highlighted biological processes like 'lipid metabolic process', 'acyl-CoA biosynthetic process', 'fatty acid elongation' and 'triglyceride catabolic process' (*Supplementary file 2*). Similar GO categories were enriched among the 154 antagonistically evolving genes that were upregulated in males but downregulated in females (*Supplementary file 2*). Interestingly, two apparently counteracting processes, fatty acid synthesis and degradation, were

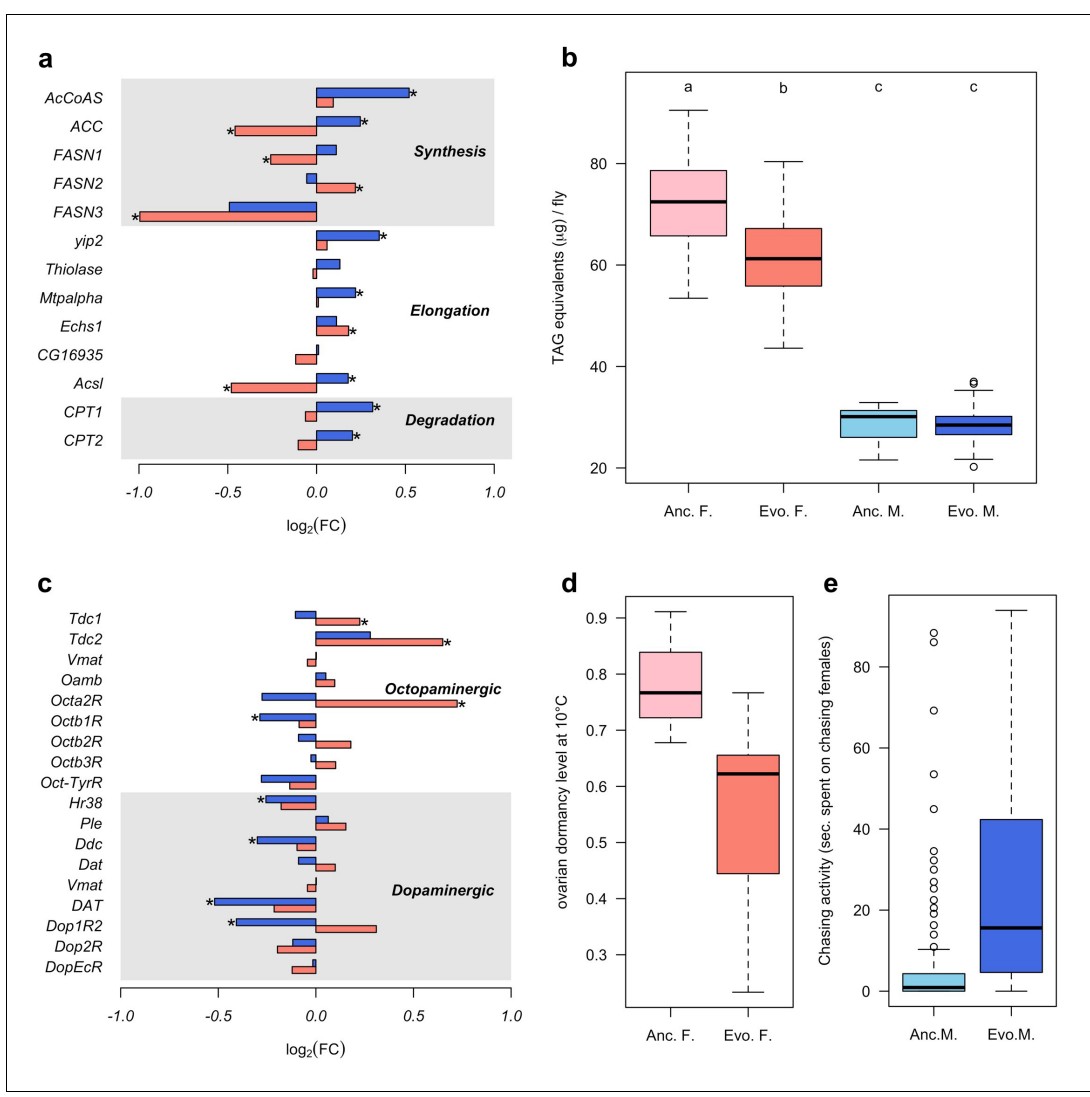

**Figure 2.** Sex-specific phenotypic evolution. (**a** and **c**) Genes involved in fatty acid metabolism and monoaminergic neural signaling evolve in response to high temperature. The evolutionary changes in males (blue bar) and females (red bar) are shown on log₂ scale. Statistical significance (FDR < 0.05) is indicated with *. For both set of genes, the evolution is largely sex-specific or even sexually discordant. (**b**) Level of triglycerides, the main constituent of body fat data from *Barghi et al. (2019)*. Evolved females have significantly lower fat content than the ancestral ones. No significant difference is found in males. Two-way ANOVA and Tuckey's HSD test. (**d**) Ovarian dormancy incidence at 10°C in ancestral and evolved females. Evolved females have a lower dormancy incidence than ancestral ones (Wilcoxon's test, W = 1.5, p=0.028). (**e**) Time males chasing females. Evolved males spent significantly more time chasing females (Wilcoxon's test, W = 1323.5, p<0.001).

The online version of this article includes the following figure supplement(s) for figure 2:

**Figure supplement 1.** Ovarian dormancy incidence at 12°C.

**Figure supplement 2.** Time male flies attempting to copulate.

both upregulated in males (*Figure 2a*) whereas in females, only genes involved in fatty acid synthesis were significantly downregulated (*Figure 2a*). A link between these changes in gene expression and a higher-level phenotype is suggested by the observation that these laboratory populations experienced a significant decrease of fat content only in females but not in males (*Barghi et al., 2019*; *Figure 2b*).

In addition, sex-specific responses to selection in gene expression were also related to neuronal signaling. The evolution of dopamine signaling during temperature adaptation has previously been reported in male flies of the same population (*Jakšić et al., 2019*). The 1086 genes that evolved decreased expression in males were enriched in brain and ganglion tissues (FET, FDR < 0.001 in both tests; *Figure 1d* and *Supplementary file 1*) whereas there was no enrichment in these tissues for females. Likewise, gene expression of dopaminergic processes (e.g.: *Ddc*, *DAT* and *Dop1R2*) evolved downregulation in males but did not evolve in females (*Figure 2c*). In contrast, only females evolved increased expression of genes involved in octopamine biosynthesis and signaling (e.g.: *Tdc1*, *Tdc2* and *Octα2R*) (*Figure 2c*).

The sex-specific modulation of transcriptional activity in different neuronal circuits may trigger changes in sex-specific fitness-related behaviors such as male courtship and female oviposition. In support of this hypothesis, the GO terms 'copulation' and 'male courtship behavior' were enriched among the 154 antagonistic genes up-regulated in males, as was 'oviposition' among the 1877 genes with female-specific up-regulation (*Supplementary file 2*). The increased fecundity of evolved females (*Barghi et al., 2019*) fits the expectations for increased octopamine synthesis (*Cole et al., 2005*; *Monastirioti, 2003*). Female fecundity is, however, a complex trait which may be affected by many factors other than increased octopamine level. We tested therefore another octopamine-related phenotype that was not selected in the experiment, ovarian dormancy in response to cold temperatures (*Andreatta et al., 2018*). Confirming the increased octopamine level in the evolved females, dormancy incidence was lower at two different dormancy-inducing temperatures (10°C and 12°C) (*Figure 2d* and *Figure 2—figure supplement 1*). Further, we also observed changes in male-specific behavior after 100 generations of adaptation; evolved males spent more time chasing females and made more copulation attempts than ancestral ones (*Figure 2e* and *Figure 2—figure supplement 2*).

The sexually discordant evolution of several phenotypes, including gene expression, metabolism and behavior, provides evidence that sex-specific adaptive processes occurred in experimental populations exposed to a novel temperature regime. This raises the important question of how potentially conflicting selection pressures on the shared genome have been decoupled during 100 generations of evolution.

## Rapid sex-specific adaptation can be driven by altered sex-biased gene regulation

Sexually dimorphic gene expression is abundant in *Drosophila* (*Parsch and Ellegren, 2013*) and 95% of the genes in the ancestral population of this study are also sex-biased (*Supplementary file 1*). This implies the decoupling of selection on the gene expression in males and females (*Mank, 2017a*) as well as the presence of a sex-biased regulatory architecture of the transcriptome (*Mank, 2017b*; *Parsch and Ellegren, 2013*; *Pennell and Morrow, 2013*) in the ancestral population. Transcription factors (TF) with sex bias in expression or splicing are the key factor underlying this sex-biased regulatory architecture (*Mank, 2017a*). It has been hypothesized that relatively fast sex-specific responses to discordant selection may be driven by fixation of novel mutations resulting in sex-biased gene expression (*Stewart et al., 2010*; *van Doorn, 2009*). However, we observe sex-specific expression changes across replicates after only 100 generations, in which case de novo mutations in individual replicates are unlikely to be the driver (*Burke et al., 2010*). Rather, selection on standing genetic variation in existing sex-specific genetic architecture seems the most likely mechanism allowing replicated populations to approach different functional requirements of the two sexes in the new environment over such a short timescale (*Figure 3*).

Candidate TFs supporting this hypothesis would regulate both genes with sex-biased expression (criterion 1) and genes with a significant evolution of sex bias in expression (criterion 2). Furthermore, the sex bias of these TFs must have evolved in a direction compatible with the changes of their target genes (criterion 3). Of 656 annotated TFs expressed in our populations, 300 TFs evolved a change in sex-biased expression (i.e. either evolve a new sex bias or the ancestral sex bias changes);

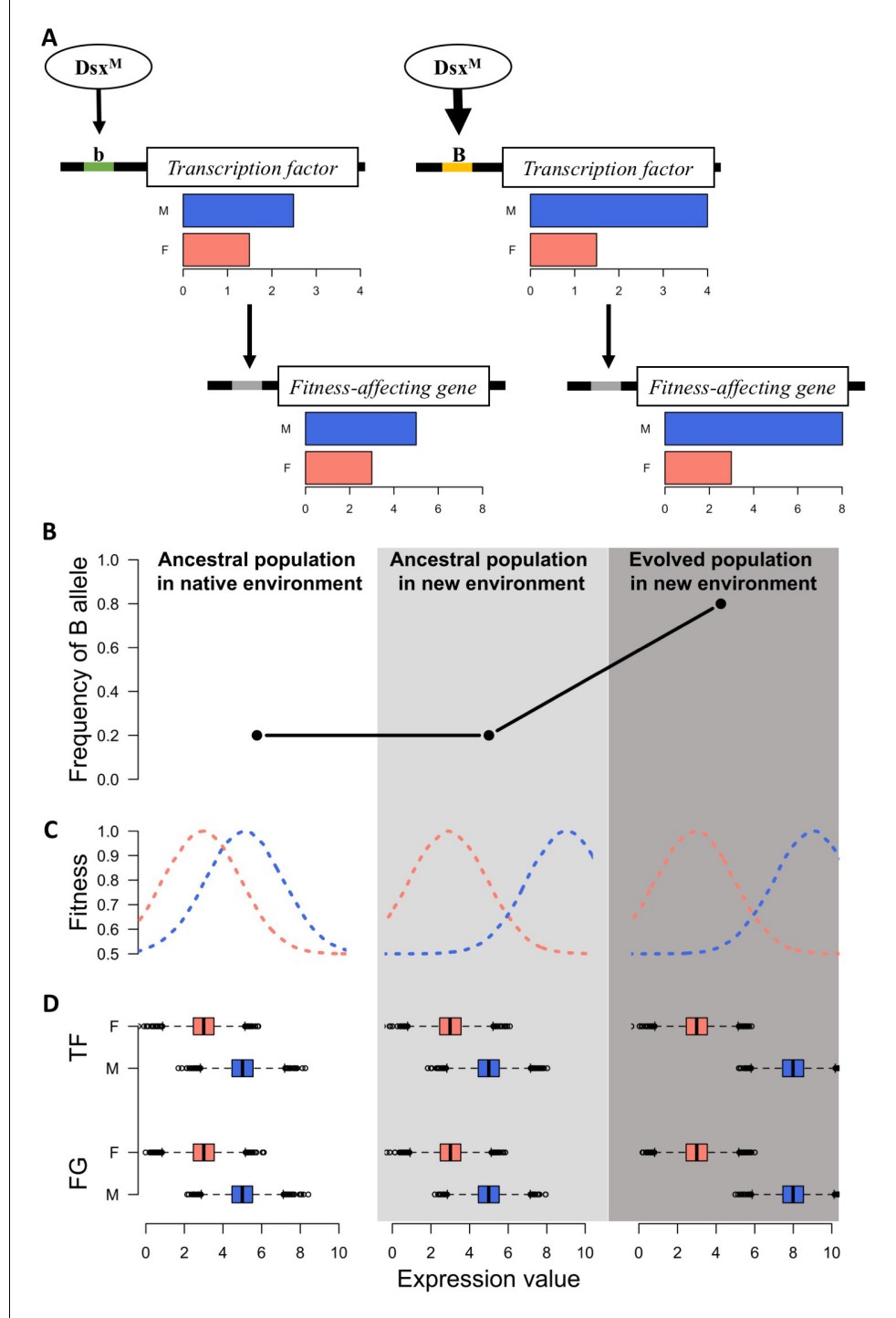

**Figure 3.** A simple model for rapid evolution of sex-specific adaptation. Regulatory variation segregating at a transcription factor is selected for a more pronounced difference in gene expression between sexes. This also causes more pronounced expression differences in a downstream gene satisfying the altered requirements of the two sexes in the new environment. (**A**) Regulatory cascade of a transcription factor (TF) controlled by sex-specific isoforms of Dsx. Two alleles with different binding affinity (B > b) with DsxM but not with DsxF are regulating downstream genes affecting fitness (FG). (**B**) Frequency of the allele increasing sex bias (B allele) at three different stages: in the native (natural) environment, in the new hot environment at the start of the experiment, in the new hot environment at the end of the experiment. (**C**) Fitness landscape at the three different stages. (**D**) Expression
*Figure 3 continued on next page*

*Figure 3 continued*

of TF and FG in males and females at the different stages. After 100 generations, the frequency increase of the allele increasing sex-biased expression of the TF results in a resolved intra-locus conflict.

210 and 80 evolved either in females or males, respectively, and 10 changed in opposite direction in the two sexes (*Supplementary file 4*). Based on cis-regulatory element enrichment, we identified 69 TFs which regulate genes with sex-biased expression and a total of 198 TFs that target genes with sex bias evolving in opposite direction (*Supplementary file 5*). In the end, 19 TFs satisfied all our three criteria for the most likely candidates targeted by the discordant selection in the two sexes (*Supplementary file 6*). Despite genomic time series data being available for these populations (*Barghi et al., 2019*), extensive linkage structure in the populations preclude an unambiguous identification of selected TF alleles. Future functional studies will show which of these candidate TFs are accomplishing the decoupling of male and female requirements and which molecular processes contribute to adaptation of the two sexes in a novel temperature regime. Nevertheless, we caution that the evolution of gene expression is most likely polygenic, with several-or even many loci contributing to the evolution of sex bias. In this case, both genomic responses and functional tests may be complicated due to the expected small effects of individual loci.

Using computer simulations, we further corroborated the hypothesis that sex-specific adaptation can be achieved rapidly in the presence of segregating regulatory variants which alter the sex bias of a trait. Based on the haplotype information of the founder lines initiating the experiment (*Barghi et al., 2019*), we simulated traits (expression value) each controlled by 50 additive loci (TFs) of which 0, 1, 2, 5, 10 or 20 are sex-specific (effect size = 0 in one sex)/sex-biased (2-fold difference in effect size). The simulated populations were exposed to a selection regime where males and females of the same population have different fitness optima for the focal trait and we monitored the phenotypic change in each sex during 100 generations. 100 simulations were performed under each scenario. Without sex bias in the effect size ($r_{mf}$ = 1), neither males nor females could respond to the discordant selection (*Figure 4*). With 40% of the loci contributing to the trait being sex-specific ($r_{mf}$ = 0.49 ± 0.2) or sex-biased ($r_{mf}$ = 0.87 ± 0.05), both males and females can evolve toward the opposing optima (*Figure 4* and *Figure 4—figure supplement 1*). Nevertheless, sex-specific or sex-biased expression is not required for many contributing loci. Already two sex-specific ($r_{mf}$ = 0.94 ± 0.08) loci significantly decouple the response of the two sexes (*Figure 4b*) under opposing selection pressures.

## Maintenance of genetic variation with sex-biased effects

As discussed above, the rapid sex-specific responses, which are highly parallel across replicates, in combination with the gain and loss of sexual dimorphism (*Figure 1—figure supplement 2*) highlight the importance of standing genetic variation in sex-biased regulatory architecture. This raises the interesting question of how genetic variation with sex-biased effects is maintained. Assuming a simple genetic basis and a stable fitness landscape with pronounced differences between the two sexes, alleles with dimorphic expression are expected to become fixed. We propose two, not mutually exclusive hypotheses to explain the discrepancy to our observation. First, the fitness landscape of some sex-specific phenotypes could vary in response to environmental fluctuation. In this case, alleles controlling the sex difference of a trait could be segregating and maintained in a population. As natural *Drosophila* populations regularly encounter seasonal temperature fluctuations, candidate alleles regulating sex-specific temperature adaptation can be maintained at sufficiently high frequencies to facilitate rapid responses. The impact of seasonal variation on oscillating allele frequency changes has been recently described experimentally and theoretically (*Bergland et al., 2014*; *Wittmann et al., 2017*). The second hypothesis assumes a polygenic basis. We note that unambiguous sex-limited modifiers (e.g. male and female isoforms of *doublesex* ; *Kopp et al., 2000*) do not preclude polygenic adaptation — these sex-limited modifiers may regulate many down-stream regulators that respond to the environmental change. Thus, already minor frequency shifts of these down-stream regulators could mediate the observed evolution of sex-specific gene expression changes. Importantly, under polygenic adaptation segregating variation is maintained for rather long time-scales (*Barton and Keightley, 2002*; *Gillespie, 1984*; *Gillespie and Turelli, 1989*).

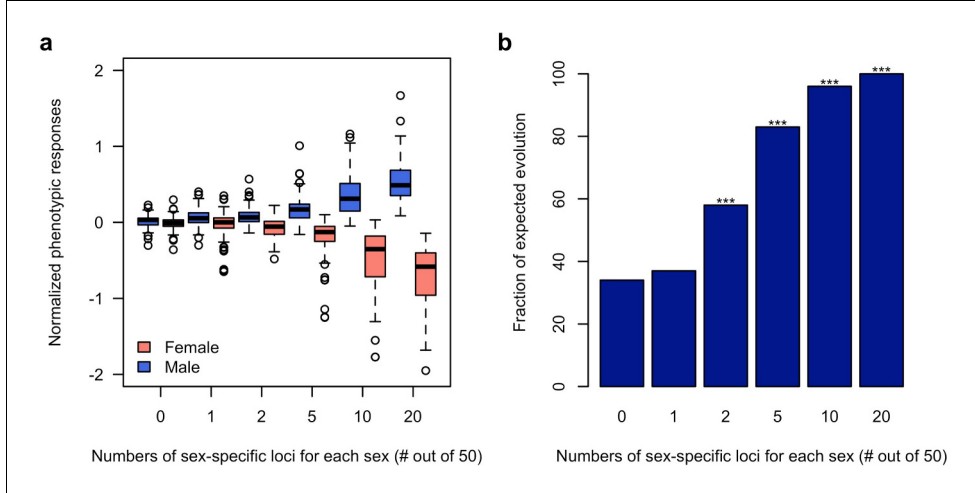

**Figure 4.** Rapid decoupling of the phenotypic response to sexually discordant trait optima by a few sex-specific loci. (a) The phenotypic response of a trait controlled by 50 loci after 100 generations of sexually discordant selection. Different numbers of sex-specific loci in each sex are shown. For each scenario, 100 independent computer simulations were performed. The normalized phenotypic change is calculated as the ratio between phenotypic change and phenotypic variance of the ancestral population. (b) Fraction of simulations for which the focal trait increases in males but decreases in females. The statistical significance denoted by an asterisk is based on one-sample proportion test comparing to the control simulation without any sex-specific locus. Bonferroni's correction is applied. Already two sex-specific loci in each sex significantly decouples the phenotypic responses to the discordant selection. With increasing numbers of sex-specific loci, the difference between the sex-specific phenotypic responses becomes more pronounced.
The online version of this article includes the following figure supplement(s) for figure 4:

**Figure supplement 1.** Sex-specific responses to discordant selection via sex-biased loci.

Indirect support for this hypothesis comes from the observation that no significant SNPs explaining the sex difference for multiple human traits can be identified (*Randall et al., 2013*). Under this scenario, rapid evolution of the sex difference may be achieved by the heterogeneous genotypic changes across replicated populations (*Barghi et al., 2019*).

## Conclusion and outlooks

This study demonstrates the power of experimental evolution to study sex-specific adaptation after an environmental shift. A substantial fraction of the transcriptome and related high-level phenotypes rapidly developed discordant changes in the two sexes upon exposure to a new environment. We propose that variation segregating in the ancestral population has facilitated the evolution of sex-biased gene expression, which in turn provides the basis for the sex-specific adaptation evidenced by the broad range of phenotypes evolving in different directions in males and females.

While we provided robust evidence for sex-specific adaptation, it is important to keep in mind that the identification of the selected traits in both sexes is an extremely challenging task. While 60% of genes changed expression in a sex-specific manner, it is unlikely that each of them is independently selected. We can anticipate many ways how the sex specific phenotypic changes have been achieved, ranging from allometric changes during adaptation to selection acting on cis-regulatory variation of highly pleiotropic transcription factors. Further characterization of the adaptive changes needs to distinguish between two goals. One goal, which is pursued in many studies, is the identification of the gene(s) that experienced a frequency change of a favored variant as contribution to the adaptive phenotype. The other goal is the identification of the selected phenotype. Given the pleiotropic effects of many genes and the polygenicity of most adaptive phenotypes (*Barghi et al., 2019*; *Pritchard et al., 2010*), it is apparent that the characterization of individual selected alleles has clear limitations in reaching the later goal. In fact, the justification of studies aiming to characterize adaptive allelic variants has been challenged (*Rockman, 2012*). More rewarding would be the

characterization of the adaptive trait, which is selected in a sex-specific manner. Our enrichment analysis and characterization of high-level phenotypes aimed towards this direction, but we cannot distinguish between correlated phenotypic changes and the actual selected phenotypes.

While most of this report focused on the rapid evolution of sex-specific adaptation, the driving forces behind this have not been discussed to the same extent, largely because they will require further functional characterization. Nevertheless, in line with sex-dependent dietary effects on fitness (*Camus et al., 2019*), the fact that males and females have vastly different functional requirements after being exposed to a novel environment has far reaching consequences-well beyond *Drosophila*. We anticipate that our results will have profound influence on biomedical research and medical treatments which need to account for the overwhelming differences of the two sexes in particular with respect to new environmental stressors, reaching from diet to climatic conditions.

## Materials and methods

### Experimental evolution

The set-up of the experimental evolution populations is described in *Barghi et al. (2019)*. In brief, 10 replicated outbred populations were constituted from 202 isofemale lines derived from a natural *Drosophila simulans* population collected in Tallahassee, Florida, USA in 2010. Replicated populations have been independently adapting to a laboratory environment at 18/28°C with 12 hr dark/12 hr light photoperiod for more than 160 generations with a census population size of 1000–1250 adults per population per generation.

### RNA-Seq common garden experiment

The collection of samples for RNA-Seq and all other phenotypic assays, was preceded by two generations of common garden rearing. Two different RNA-Seq data sets were generated for this study: The first one, in which highly replicated whole body samples were collected, represents the main dataset that we used to contrast gene expression levels of females and males from ancestral and hot evolved populations. The second one with gonads and carcass being analyzed separately was generated to correct for allometric differences between evolved and ancestral populations.

The first data set comes from a common garden experiment (CGE) performed after 103 generations of evolution in the hot environment and this CGE has been described in *Barghi et al. (2019)*; *Hsu et al. (2019)*; *Jakšić et al. (2019)*. We reconstituted five replicates of the ancestral population from 184 founder isofemale lines by generating five pools with five mated females from each isofemale line. No significant allele frequency differences are expected between the reconstituted ancestral populations and the original ancestral populations initiating the experiment (*Nouhaud et al., 2016*). Because we evaluated phenotypes on the population level, even deleterious mutations will have a very limited impact, because they occur only in a single isofemale line, which represents a very small fraction of the total population. For each of the 10 hot evolved replicates, we generated three sub-replicates by multiple egg lays. The five ancestral replicates and all hot evolved sub-replicates were reared in common garden for two generations with controlled low egg density (400 eggs/bottle) in the same temperature regime as during the evolution experiment. After two generations under CGE conditions, flies were collected from each replicate/sub-replicate a few hours after eclosion and maintained on fresh food under the 18/28°C temperature regime to allow for mating. On the third day after eclosion, sexes were separated under $CO_2$ anesthesia and allowed to recover for two days. At the age of five days, 50 flies of each sex were snap frozen in liquid nitrogen at 2pm and stored at −80°C until RNA extraction. We sequenced the transcriptomes of 50 females and males from each of the five ancestral replicates and from each of the 10 hot evolved replicates with three sub-replicates each for males and two sub-replicates for females. The third sub-replicate of the hot evolved female samples was frozen at a different age which prevented the joint analysis in the context of this study (*Hsu et al., 2019*).

The second RNA-Seq data set was generated at generation 140 of the hot evolving populations to correct for potential differences in the relative size of gonadal and carcass tissue between ancestral and evolved populations. CGE set-up and maintenance were repeated as described above, without sub-replication of the hot evolved replicates: 50 whole body samples for females and males were collected from five reconstituted ancestral and all 10 hot evolved replicates and snap-frozen at

the age of five days at 2pm. Gonadal and carcass tissue was sampled from six reconstituted ancestral and six randomly chosen hot evolved replicates (replicates no. 1, 4, 5, 6, 8, 9). For each replicate, 50 female and 50 male flies were dissected in PBS at the age of 5 days and dissected gonadal tissues and remaining carcasses were immediately preserved in Qiazol and stored at −80°C.

## RNA extraction and library preparation

Total RNA was extracted using the same procedure for all samples: homogenized in Qiazol with a pestle. Total RNA was extracted from the homogenate using the Qiagen RNeasy Universal Plus Mini kit (Qiagen, Hilden, Germany) with DNase treatment to remove traces of genomic DNA. Libraries were prepared on the Neoprep Library Prep System (Illumina, San Diego, USA) starting from 100 ng total RNA and following the manufacturer's recommended protocol for the TruSeq stranded mRNA Library Prep Kit for Neoprep. Neoprep runs were performed using software version 1.1.0.8 and protocol version 1.1.7.6 with default settings for 15 PCR cycles and an insert size of 200 bp. Libraries were arranged in randomized order on library cards. To avoid batch effects, we used library cards with the same lot number for all samples for which direct comparisons of expression levels were planned (lot no. 20123465: CGE at generation 103, males, whole body, all ancestral and hot evolved samples; lot no. 20173962: CGE at generation 103, females, whole body, all ancestral and hot evolved samples; lot no. 20182049: CGE at generation 140, females and males, whole body and gonadal tissue). 50 bp single-end reads were sequenced on an Illumina HiSeq 2500.

## RNA-Seq data processing

All sequencing reads were trimmed with ReadTools (Version: 1.5.2) (*Gómez-Sánchez and Schlötterer, 2018*) based on a quality score of 20, and mapped with GSNAP (Version: 2018-03-25; Parameters: -k 15 N 1 m 0.08) (*Wu and Nacu, 2010*) to *Drosophila simulans* reference genome (*Palmieri et al., 2015* Supplementary file 7). Exon-aligned reads were counted with Rsubread (Version: 1.30.9) (*Liao et al., 2013*) based on the annotation (*Palmieri et al., 2015*) and the expression level of each gene was quantified after normalizing the exon-aligned read counts by TMM method implemented in edgeR (Version: 3.22.5) (*Robinson et al., 2010*). Only genes with more than 0.1 count per million base pairs in each sample of the main dataset (1st CGE) were retained for the analysis to avoid biased analyses.

## Estimation and correction of the allometric difference

Using an independent CGE that consisted of dissected samples (2nd CGE, correcting dataset), we corrected for potential differences in the relative size of gonadal and remaining carcass tissues in ancestral and hot evolved populations for each gene.

For each gene, we formulated its average expression across whole-body samples ($\overline{y_{Wb, i}}$) with the average expression across gonad samples ($\overline{y_{g, i}}$) and carcass samples ($\overline{y_{c, i}}$) as: $\overline{y_{Wb, i}} = \alpha_i \overline{y_{g, i}} + (1 - \alpha_i)\overline{y_{c, i}}$, where $\alpha$ is the coefficient measuring the relative portion of gonadal expression of a gene in whole body expression, ranging from 0 to 1 (*Supplementary file 10*). If a gene is expressed at similar level in both gonadal and somatic tissues, it would not be affected by differences in tissue scaling. We excluded these genes in the comparison of tissue-scaling and applied no correction for them in the subsequent analysis. Leave-one-out cross validation was performed to evaluate the accuracy and robustness of the method. The estimation of the scaling coefficients for each gene was robust (*Supplementary file 8*). In addition, the prediction was nearly perfect (*Supplementary file 9*).

Comparing the distribution of gene-wise estimates of scaling coefficients, we found significant difference between ancestral and evolved populations for both sexes (Kolmogorov-Smirnov test D = 0.18 and 0.12 for females and males, respectively; p<0.001 in both tests; *Supplementary file 11*). This suggested that the gonad-carcass size ratio may have significantly changed during the adaptation to the new environment. A proper correction is necessary for unbiased inference. Hence, we normalized the tissue-scales of each ancestral sample to the scale of evolved samples. We reconstructed pseudo whole-body samples using the expression data of dissected samples of the ancestral populations and scaling coefficients estimated from the evolved samples as: $y_{Wb, i}^{pseudo} = \alpha_i^{evo} y_{g, i} + (1 - \alpha_i^{evo})y_{c, i}$.

Finally, the ratio of expression levels between the reconstructed pseudo whole-body samples and the original ones ($\frac{y^{pseudo}_{Wb, i}}{y_{Wb, i}}$) for each gene were calculated as the correcting factors ($\gamma_i$). Gene-wise correction was applied to ancestral whole-body samples from the 1$^{st}$ CGE by multiplying the expression value of each gene to corresponding $\gamma_i$. The corrected samples were used in all subsequent analyses.

## Differential expression (DE) analysis

After correction, we modeled the effects of sex and evolution on gene expression variation as: $Y = group + \varepsilon$, where $Y$ is the normalized expression values; $group$ indicates the combination of evolution and sex difference with four levels (ancestral females, ancestral males, evolved females and evolved males) and $\varepsilon$ is the random error. Likelihood ratio tests implemented in edgeR were used to perform differential expression analysis on three contrasts: (1) *female evolution*: evolved females vs. ancestral females, (2) *male evolution*: evolved males vs. ancestral males and (3) *sex bias*: females vs. males. Benjamini and Horchberg's FDR correction (*Benjamini and Hochberg, 1995*) was applied with the significance threshold of FDR < 0.05. Genes showing distinct evolutionary patterns were classified based on criteria in (*Supplementary file 2*).

## Enrichment analysis

Gene ontology (GO) enrichment was performed using the default 'weight01' algorithm implemented in topGO (version 2.32.0) (*Alexa et al., 2006*). Genes highly expressed in each tissue were identified based on the FlyAtlas expression dataset (*Chintapalli et al., 2007*) (required >2 fold higher expression in a certain tissue than whole-body, *Supplementary file 3*). Fisher's exact test was applied for the enrichment of tissue expression. Except for the GO enrichment analysis of which the method already accounts for multiple testing (*Alexa and Rahnenführer, 2018*), Benjamini and Horchberg's FDR correction (*Benjamini and Hochberg, 1995*) was applied to account for multiple testing.

## Cis-regulatory element enrichment analysis

Enrichment of cis-regulatory elements (CREs) 5 kb upstream and intronic sequences of the genes of interest (*Supplementary file 5*) was identified with RcisTarget (version 1.0.2) (*Aibar et al., 2017*). We searched for enriched motifs using the latest motif ranking file of *Drosophila* species ('dm6-5kb-upstream-full-tx-11species.mc8nr.feather', accessed on 2019-04-08) with parameters, nesThreshold = 3 and aucMaxRank = 1%. Transcription factors (TFs) annotated to bind on the enriched CREs were considered as candidate master TFs regulating the genes of interest.

We performed cis-regulatory element enrichment analysis on female-biased, male-biased, female-specifically up-regulated, down-regulated, male-specifically up-regulated, down-regulated, and two sets of antagonistically evolving genes separately (*Supplementary file 5*).

## Male reproductive activity assays

We measured the reproductive activity of five reconstituted ancestral populations and five randomly selected hot evolved replicates at generation 140. After two generations reared in a common garden condition (18/28℃ cycling), 10 five-day-old mated males and females from each population were placed together in an agar-based arena (4% agar, 4% sugar) and filmed for 15 min at 20 FPS (frame-per-second) at 28℃ using the FlyCapture2 system (PointGrey, Version 2.13.3.31). In total, 10 video each for reconstituted ancestral and evolved populations were filmed. The movement and behavior of each fly was tracked using flytracker (Version 1.0.5) (*Eyjolfsdottir et al., 2014*). Videos that failed the tracking process were not used for subsequent analysis. Janelia Automatic Animal Behavior Annotator (JAABA, Version 0.6.0_2014a) was used to annotate and recognize the chasing and attempted copulation behavior (*Kabra et al., 2013*). We imported the output files of JAABA into R for data processing and statistical analysis. The time a male fly spent on chasing and copulation attempt females was quantified. Wilcoxon's rank sum test was applied to test the difference in reproductive activity of male flies in evolved and ancestral populations.

## Female reproductive dormancy assays

We screened three replicates of the reconstituted ancestral and 10 replicated evolved populations for dormancy incidence at generation 167. Ancestral and evolved populations were kept at the same

temperature regime for four generations before freshly eclosed female flies were collected within two hours post-eclosion and kept under dormancy-inducing conditions (10°C and 12°C, LD 10:14) for three weeks before dissection. 90 flies from each population and temperature regime were dissected and their oogenesis progression was examined. Each fly was classified as dormant or non-dormant (*Lirakis et al., 2018*). Wilcoxon's rank sum test was applied to test the difference in dormancy level of female flies in evolved and ancestral populations.

### Simulation

We performed forward simulations using *qff* function implemented in MimicrEE2 (v208) (*Vlachos and Kofler, 2018*). Starting with 189 founder haplotypes (*Barghi et al., 2019*), in each sex, we simulated a trait controlled by a varied number of loci (0, 1, 2, 5, 10, 20) conferring sex-specific or sex-biased effects while the total number of contributing loci in each sex was constantly 50. For each trait, we assumed an additive model ($a \sim \Gamma(0.5, 2.5)$) and relatively high heritability ($h^2 = 0.8$). A sex-specific locus confers additive effect on a trait in one sex but no effect in the other sex while a sex-biased locus is assumed to contribute to the trait in both sexes but there is a 2-fold difference in its additive effect between the two sexes. Sexually discordant selection, where the trait optimum is shifted three units (i.e. on average, 1.9 phenotypic standard deviations) to the left and to the right for males and females respectively, was imposed to the simulated traits for 100 generations assuming balanced sex-ratio. In total, we performed 100 independent simulations for each of the six scenarios in this study. Then, we measured the normalized phenotypic responses to the selection as $\frac{\Delta \overline{p}_{100-0}}{\sigma_0^2}$, where $\Delta \overline{p}_{100-0}$ is the mean phenotypic difference between F100 and F0 of the populations and $\sigma_0^2$ is the phenotypic variance when the experiment starts. We calculated the fractions of simulations in which expected phenotypic responses in the two sexes (increase in males but decrease in females) were observed. One-sample proportion test was performed to test for significant difference between each scenario to the control group. Bonferroni's correction was applied to account for multiple testing.

## Acknowledgements

We thank Ray Tobler, François Mallard, Kathrin A Otte and Felix Lehner for assistance during the experiments, and Scott Allen for comments on earlier draft of the manuscript. Illumina sequencing was done by the Vienna Biocenter Core Facilities (VBCF) NGS Unit. This work was supported by European Research Council (ERC, ArchAdapt), and Austrian Science Funds (FWF, W1225).

## Additional information

### Funding

| Funder | Grant reference number | Author |
| --- | --- | --- |
| European Research Council | ArchAdapt | Christian Schlötterer |
| Austrian Science Fund | W1225 | Christian Schlötterer |

The funders had no role in study design, data collection and interpretation, or the decision to submit the work for publication.

### Author contributions

Sheng-Kai Hsu, Conceptualization, Formal analysis, Investigation, Visualization, Writing - original draft; Ana Marija Jakšić, Conceptualization, Investigation, Writing - review and editing; Viola Nolte, Manolis Lirakis, Investigation, Methodology, Writing - review and editing; Robert Kofler, Software, Methodology, Writing - review and editing; Neda Barghi, Investigation, Writing - review and editing; Elisabetta Versace, Resources, Methodology; Christian Schlötterer, Conceptualization, Supervision, Funding acquisition, Writing - original draft

**Author ORCIDs**
Sheng-Kai Hsu (iD) https://orcid.org/0000-0002-6942-7163
Ana Marija Jakšić (iD) https://orcid.org/0000-0002-2293-255X
Neda Barghi (iD) https://orcid.org/0000-0003-3700-0971
Elisabetta Versace (iD) https://orcid.org/0000-0003-4578-1851
Christian Schlötterer (iD) https://orcid.org/0000-0003-4710-6526

**Decision letter and Author response**
Decision letter https://doi.org/10.7554/eLife.53237.sa1
Author response https://doi.org/10.7554/eLife.53237.sa2

## Additional files

**Supplementary files**

• Supplementary file 1. Likelihood ratio test for different contrasts. The file records the design matrix and results of likelihood ratio test for three different contrasts between (1) evolved and ancestral samples in males, (2) evolved and ancestral samples in females and (3) male and female samples.

• Supplementary file 2. Gene ontology (GO) enrichment analysis on genes of interest. Results of gene ontology enrichment analysis using topGO among different sets of genes showing distinct expression changes were shown.

• Supplementary file 3. Enrichment analysis of genes highly expressed in each tissue among the genes of interest. In this file, we reported the results of Fisher's exact test for enrichment of genes highly expressed in each tissue among the genes of interest and the list of genes that are highly expressed in each tissue.

• Supplementary file 4. All expressed TFs annotated by RcisTarget and their evolutionary patterns. The genomic position, sex-specific evolutionary pattern and gene description of all expressed transcription factors (TFs) annotated by RcisTarget are shown.

• Supplementary file 5. Enrichment of cis-regulatory elements and identification of putative TFs among genes of interest. The outputs of RcisTarget that test for enrichment of cis-regulatory elements among each set of genes of interest are shown. The identities, expression patterns, and functional descriptions of the TFs that putatively regulates genes of interest are summarized.

• Supplementary file 6. TFs that regulate sex-biased expression, expression evolution and showed significant evolution in their expression. A list of candidate TFs that satisfy the three criteria supporting the hypothesis that selection on sex-biased transcription factors may facilitate rapid sex-specific evolution in gene expression.

• Supplementary file 7. Mapping statistics. Mapping statistics of all the samples involved in the tests in this study are reported.

• Supplementary file 8. Robustness of the estimation of allometric coefficients ($\alpha$). The Robustness of the estimation was evaluated with Jackknife sampling. The correlation of the estimates between each pair of Jackknife samples are reported.

• Supplementary file 9. Prediction accuracy of the whole body expression using the estimated allometric coefficients ($\alpha$) and the expression profiles in dissected samples. For each Jackknife sampling, the estimated allometric coefficients ($\alpha$) were applied to predict the whole body expression of the left-out sample based on its expression profiles in gonad and carcass. Pearson's correlation between the true and predicted values were reported.

• Supplementary file 10. Numeric example for the allometric estimation. An allometric estimate ($\alpha_i$) measures the abundance of a gene in gonads relative to the overall (mean) abundance in the whole body, reflecting the relative size of gonad in whole body. It may differ between populations. Genes with different expression levels in each tissue (gene1 in the figure) would be affected and thus are informative for the estimation. However, for genes with similar expression in different tissues (gene2 in the figure), they would be affected and the estimation of $\alpha_i$ would be meaningless.

• Supplementary file 11. Allometric estimate of gonadal tissues in whole bodies of each gene. An allmoetric estimate ($\alpha_i$) is the coefficient measuring the abundance of a gene in gonad relative to the

overall abundance in the whole body. The distributions of the estimates differ significantly between evolved and ancestral populations in both sexes (Kolmogorov-Smirnov test, D = 0.18 and 0.12 for females and males, respectively; p<0.001 in both tests).

- Transparent reporting form

## Data availability

Sequencing reads have been deposited in European Nucleotide Archive (ENA) under the study accession number PRJEB35504 and PRJEB35506. Original data for each plot could be found as supplementary files or in the github repository of this study (https://github.com/ShengKaiHsu/Dsim_sex-specific_adaptation; copy archived at https://github.com/elifesciences-publications/Dsim_sex-specific_adaptation).

The following datasets were generated:

| Author(s) | Year | Dataset title | Dataset URL | Database and Identifier |
|---|---|---|---|---|
| Jakšić AM, Karner J, Nolte V, Hsu S-K, Barghi N, François M, Otte KA, Lidija S, Senti K-A, Schlötterer C | 2020 | RNA-Seq data for Neuronal function and dopamine signaling and Rapid sex-specific adaptation | https://www.ebi.ac.uk/ena/data/view/PRJEB35504 | European Nucleotide Archive, PRJEB35504 |
| Hsu S-K, Jakšić AM, Nolte V, Lirakis M, Kofler R, Barghi N, Versace E, Schlötterer C | 2020 | RNA-Seq data for Rapid sex-specific adaptation | https://www.ebi.ac.uk/ena/data/view/PRJEB35506 | European Nucleotide Archive, PRJEB35506 |

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
