## [Decision Letter]

**Acceptance summary:**

In times of climate change, our awareness for adaptive process dealing with changes in temperature is undergoing a renaissance. This study demonstrates nicely the value and power of experimental evolution to study adaptation. The demonstration of sex-specific adaptation is novel and interesting for the readers of *eLife*.

**Decision letter after peer review:**

Thank you for submitting your article entitled "Rapid sex-specific adaptation to high temperature in *Drosophila*" for consideration by *eLife*. Your article has been reviewed by three reviewers, and the evaluation has been overseen by Diethard Tautz as the Senior Editor, and a Reviewing Editor.

The reviews were generally positive, but there are some issues that need to be addressed before acceptance, as outlined below:

With climate change influencing environmental conditions worldwide, adaptation to temperature has become a topic of high interest. The current study asks how do flies adapt to a novel temperature. For this, *Drosophila* simulans populations were kept for 100 generations under novel temperature conditions, all adapting in parallel. The studies focusses on experimental evolution of sex-specific gene expression. For this, the authors analyzed transcriptomic responses of males and females *Drosophila* simulans to a regime of elevated temperature. They showed that both sexes evolved differently, with many genes showing sex-specific expression evolution, and a small quantity of genes either showing congruent or opposite evolution. This evolution would be due to selection that acted on standing genetic variation rather than de novo mutations. The authors examine males and females separately and find that the two sexes evolve almost completely independently. That is, most genes change expression in only one sex, or change in opposite ways in the two sexes. This suggests that male and female gene expression are largely decoupled, and that there was a considerable amount of polymorphism with sex-specific effects segregating in the ancestral population. The genes changing expression fall into functional groups that appear to be relevant for sex-specific adaptation and the authors show that some sex-specific phenotypes have changed in the evolved populations. Finally, a theoretical model adds to the quality of the study.

The results of this study are new and will come as a surprise for many people, as it is a priori not expected that selection by temperature would be sex specific. All three reviewers agree that the study is sound and interesting, but needs some changes to be publishable in *eLife*. Below is a list of comments summarising the details of the three reviewers. A recurrent theme is that the methods are not well enough explained and need clarification. Furthermore, the consequences of the specific experimental design need clarification and placement in a proper context (e.g. the setting up of the starting populations).

Issues that need to be addressed before acceptance (combined and restructured from the three reviewers):

If I understand the experimental design correctly, the ancestral population was reconstituted from isofemale lines after the experiment of Bardhi et al., 2019 finished, and as such might not have the same history as the ancestral populations used in Barghi et al., 2019. Although surely minor, there might be transcriptomic differences between the ancestral population and the evolved populations that are not linked to the temperature regime (i.e. variation in lab environment of the isofemale lines that occurred during the Barghi experiment, creating a difference between the two "ancestral" populations). A comparison of the ancestral population from this manuscript and that of the paper in Bardhi et al., 2019 (population before temperature exposure) might have been of interest for controlling for such differences. I do not believe this impacts the results of the present study, but I think the authors could mention this peculiarity of the experimental design more in details, and maybe prove me wrong citing other studies demonstrating that transcriptomically speaking, the isofemale lines are relatively stable in time so that ancestral populations can be reconstituted at any time point?

A chief limitation of the study is that the starting populations were founded by crossing about 200 isofemale lines. As such massively hybridized populations are only rarely found in nature, the general applicability of the results are open to debate. On the other hand, this method of initializing experimental evolution does allow the crude regeneration of comparable "ancestral" controls for the purpose of subsequent assays. It is recommended to discuss these issues.

Along these lines, the artificial nature of the initial populations undergoing selection should be clearly delineated, and compared with the protocols used in other studies of *Drosophila* experimental evolution. Likewise, the five crosses used to generate the "ancestral" controls seem to have been generated with fewer isofemale lines, and thus may be subject to still greater potential problems with sampling effects, compared to the lines used to initiate the project.

In the "estimation and correction for allometric differences" section, something seems to be missing about the α coefficient. I do not understand how, if a gene has the same level of expression in the gonad and the whole-body dataset, α can either be below 0, or over 1 (since it is a comparative calculation isn't this a one option result?) I do not understand what this parameter really measures. Can you provide an example for the calculation of α and of the average expression in whole body sample in the text, if a gene has a similar level of expression, and if it has a different level of expression?

"we observe parallel sex-specific expression changes in all replicates". Is this really the case? Do the same expression changes occur in each of the 10 replicate populations? Or is it the case that mean expression across the 10 replicates is significantly higher than the ancestral population, but expression does not change in some replicates (or even changes in the opposite direction in some replicates)? From the Materials and methods section, it appears that only a significant difference was required between ancestral and evolved, not that all evolved replicates had to show the same response. This is also the impression I get from Figure 1—figure supplement 1. Please clarify.

"new mutations affecting sex bias are unlikely given a previous study" – here it would be better to briefly state why new mutations affecting sex bias are unlikely, rather than citing the reference without any other explanation. Also, the cited paper deals with yeast. Is it applicable to sex-biased expression in *Drosophila*?

Some of the Venn-like diagrams of expression patterns are relatively opaque, and should be better explained in their captions and in the general text. It is possible to guess at what is going on, but certainty is always better.

The simulations of an explicit population genetic hypothesis are a useful feature of the research. But the design of these simulations and the interpretation of the results of the simulations should be clearer. It seems as if a concern about manuscript length has led to an unfortunate degree of semantic compression, with attendant obscurity.

---

## [Author Response]

[…] Issues that need to be addressed before acceptance (combined and restructured from the three reviewers):If I understand the experimental design correctly, the ancestral population was reconstituted from isofemale lines after the experiment of Bardhi et al., 2019 finished, and as such might not have the same history as the ancestral populations used in Barghi et al., 2019. Although surely minor, there might be transcriptomic differences between the ancestral population and the evolved populations that are not linked to the temperature regime (i.e. variation in lab environment of the isofemale lines that occurred during the Barghi experiment, creating a difference between the two "ancestral" populations). A comparison of the ancestral population from this manuscript and that of the paper in Bardhi et al., 2019 (population before temperature exposure) might have been of interest for controlling for such differences. I do not believe this impacts the results of the present study, but I think the authors could mention this peculiarity of the experimental design more in details, and maybe prove me wrong citing other studies demonstrating that transcriptomically speaking, the isofemale lines are relatively stable in time so that ancestral populations can be reconstituted at any time point?A chief limitation of the study is that the starting populations were founded by crossing about 200 isofemale lines. As such massively hybridized populations are only rarely found in nature, the general applicability of the results are open to debate. On the other hand, this method of initializing experimental evolution does allow the crude regeneration of comparable "ancestral" controls for the purpose of subsequent assays. It is recommended to discuss these issues.Along these lines, the artificial nature of the initial populations undergoing selection should be clearly delineated, and compared with the protocols used in other studies of *Drosophila* experimental evolution. Likewise, the five crosses used to generate the "ancestral" controls seem to have been generated with fewer isofemale lines, and thus may be subject to still greater potential problems with sampling effects, compared to the lines used to initiate the project.

We would like to address the comments regarding the constitution and reconstitution of ancestral populations together by discussing the similarity (1) among the independent replicates of the ancestral populations and (2) between the “reconstituted” ancestral populations and the real ancestral populations initiating the evolution experiment.

Our protocol for the initiation of the experiment aims to maximally mimic the allelic composition in nature. By establishing isofemale lines we keep one random haplotype from a natural population in each isofemale line. By simply pooling all these haplotypes and allowing random mating, we are able to constitute populations with the same allele frequency spectrum as in nature to start the experiment (within the limits of sampling variation) but allow independent recombination, drift and selection response in each replicate.

Unlike evolution experiments in other organisms where freezing or seed preservation is possible, it is relatively difficult to compare the evolving *Drosophila* populations to their ancestors phenotypically. Thus, as understood by the reviewers, replicates of ancestral populations were reconstituted multiple times before each common garden phenotyping experiment. Potential adaptation to the lab environment with the residual heterogeneity or de novo mutation in each line could be possible. Nevertheless, as discussed (Barghi et al., 2019), given the small effective population size during the maintenance of each isofemale line, most mutations are effectively neutral and adaptation is unexpected. It is, however, possible that de novo deleterious mutations are accumulated through time, but every isofemale line will accumulate different de novo mutations. Because we are phenotyping on the population level, any mutation in single isofemale lines will have only a very minor effect on the population level phenotype. In the worst case, where the mutation is dominant and fixed in one isofemale line, it would only affect the phenotypic value of ~1% (2pq+q^2^, where q = 1/202) of the whole population.

Furthermore, we have shown that populations constituted from the same set of isofemale lines at different time points (several years apart) are not different from each other in terms of allele frequency along the whole genome (Nouhaud et al., 2016). This suggests the robustness of the protocol. Additional clarification has been added to the revised manuscript (subsection “RNA-Seq common garden experiment”, second paragraph).

In the "estimation and correction for allometric differences" section, something seems to be missing about the α coefficient. I do not understand how, if a gene has the same level of expression in the gonad and the whole-body dataset, α can either be below 0, or over 1 (since it is a comparative calculation isn't this a one option result?) I do not understand what this parameter really measures. Can you provide an example for the calculation of α and of the average expression in whole body sample in the text, if a gene has a similar level of expression, and if it has a different level of expression?

We understand that this analysis is convoluted and not straightforward for understanding. Corresponding changes (subsection “Estimation and correction of the allometric difference”) and a supplementary figure (Method—figure supplement 1) with numeric examples have been made in the revised manuscript for better understanding. To answer the questions to the reviewers, α in this study measures the relative portion of gonadal expression of a gene in whole body expression. Intuitively, it reflects the relative size of gonad in the whole body. Since we used the gene expression in whole body for statistical testing, a change in the relative sizes between tissues might change the expression value of a gene in whole body even when there’s no regulatory change in each tissue and leads to potential misinterpretation, as shown in the figure. Thus, it is important to estimate this parameter in each population and compare it. Genes with different expression levels in different tissues (e.g.: gene1 in the figure) would be affected and thus are informative for the estimation. However, for genes with similar expression in different tissues (gene 2 in the figure), they would be affected and the estimation of α would be meaningless. Depending on the measuring noise, the estimates can go over 1 or below 0. To avoid confusion, this sentence is modified in the revised manuscript.

"we observe parallel sex-specific expression changes in all replicates". Is this really the case? Do the same expression changes occur in each of the 10 replicate populations? Or is it the case that mean expression across the 10 replicates is significantly higher than the ancestral population, but expression does not change in some replicates (or even changes in the opposite direction in some replicates)? From the Materials and methods section, it appears that only a significant difference was required between ancestral and evolved, not that all evolved replicates had to show the same response. This is also the impression I get from Figure 1—figure supplement 1. Please clarify.

Yes, we agree with the reviewers that our statement needs slight modification (subsection “Rapid sex-specific adaptation can be driven by altered sex-biased gene regulation”, first paragraph and legend of Figure 1—figure supplement 1). It is true that based on the DE analysis we actually searched for the genes that exhibit higher mean expression across all evolved replicates than across all ancestral replicates. We wouldn’t have comparable power for statistical tests contrasting each evolution replicate to the ancestors. Nevertheless, considering the expression changes in each replicate, we note that 91% and 87% of the candidate genes in males and females in this study change their expression to the same direction in all replicates, respectively.

"new mutations affecting sex bias are unlikely given a previous study" – here it would be better to briefly state why new mutations affecting sex bias are unlikely, rather than citing the reference without any other explanation. Also, the cited paper deals with yeast. Is it applicable to sex-biased expression in *Drosophila*?

This was an overlooked mistake. By accident, we cited the wrong paper by the same author. Thanks for pointing this out. Corrected citation and more explanation have been added to the revised manuscript (subsection “Rapid sex-specific adaptation can be driven by altered sex-biased gene regulation”, first paragraph).

Some of the Venn-like diagrams of expression patterns are relatively opaque, and should be better explained in their legends and in the general text. It is possible to guess at what is going on, but certainty is always better.

More explanation has been added in the revised manuscript (Figure 1 legend).

The simulations of an explicit population genetic hypothesis are a useful feature of the research. But the design of these simulations and the interpretation of the results of the simulations should be clearer. It seems as if a concern about manuscript length has led to an unfortunate degree of semantic compression, with attendant obscurity.

More details about the simulations have been added in the revised manuscript (subsection “Rapid sex-specific adaptation can be driven by altered sex-biased gene regulation”, last paragraph).